# The HMGB1-RAGE Axis Drives the Proneural-to-Mesenchymal Transition and Aggressiveness in Glioblastoma

**DOI:** 10.3390/ijms26199352

**Published:** 2025-09-25

**Authors:** Hao-Chien Yang, Yu-Kai Su, Vijesh Kumar Yadav, Iat-Hang Fong, Heng-Wei Liu, Chien-Min Lin

**Affiliations:** 1Division of Neurosurgery, Department of Surgery, Taipei Medical University–Shuang Ho Hospital, New Taipei City 23561, Taiwan; 19118@s.tmu.edu.tw (H.-C.Y.); yukai.su@gmail.com (Y.-K.S.); 18149@s.tmu.edu.tw (I.-H.F.); 2Department of Neurology, School of Medicine, College of Medicine, Taipei Medical University, Taipei City 11031, Taiwan; 3Taipei Neuroscience Institute, Taipei Medical University, Taipei City 11031, Taiwan; 4Graduate Institute of Clinical Medicine, College of Medicine, Taipei Medical University, Taipei City 11031, Taiwan; 5Department of Medical Research & Education, Taipei Medical University–Shuang Ho Hospital, New Taipei City 23561, Taiwan; vijeshp2@gmail.com

**Keywords:** glioblastoma, proneural-to-mesenchymal transition, HMGB1-RAGE signaling, tumor stemness, metabolic reprogramming, therapy resistance

## Abstract

Glioblastoma (GBM) remains the most lethal primary brain tumor, owing to profound intratumoral heterogeneity and the limited efficacy of standard treatments. The mesenchymal (MES) molecular subtype is particularly aggressive, exhibiting heightened invasiveness, therapy resistance, and dismal patient survival compared with the proneural (PN) subtype. Emerging evidence implicates the High Mobility Group Box 1 (HMGB1) protein and its cognate receptor, the Receptor for Advanced Glycation End Products (RAGE), as drivers of malignant progression, yet their contribution to the PN-to-MES transition is incompletely defined. We integrated transcriptomic analyses of TCGA-GBM and TCGA-LGG cohorts with immunohistochemistry on in-house patient specimens. Functional studies in patient-derived and established GBM cell lines included migration and invasion assays, tumorsphere formation assays, shRNA knockdowns, and Seahorse XF metabolic profiling to interrogate the HMGB1-RAGE axis. HMGB1 and RAGE expression was markedly elevated in MES GBM tissues and cell lines. Importantly, higher HMGB1 expression correlated with shortened overall survival (*p* < 0.009). HMGB1 silencing curtailed cell motility and downregulated core epithelial-to-mesenchymal transition markers (*N*-cadherin, Snail). RAGE knockdown diminished tumorsphere formation efficiency and reduced transcription of stemness genes (OCT4), underscoring its role in sustaining tumor-initiating capacity. Metabolically, HMGB1/RAGE activation boosted both mitochondrial respiration and glycolysis, conferring the bioenergetic flexibility characteristic of MES GBM. The HMGB1-RAGE signaling axis orchestrates mesenchymal identity, invasiveness, stem cell-like properties, and metabolic reprogramming in GBM. Targeting this pathway may disrupt the PN-to-MES transition, mitigate therapeutic resistance, and ultimately improve outcomes for glioblastoma patients.

## 1. Introduction

Glioblastoma (GBM)—now designated as glioblastoma, IDH-wildtype, CNS WHO grade 4 under the 2021 WHO Classification of Central Nervous System Tumours—is the most aggressive primary brain tumor in adults, characterized by rapid growth, diffuse infiltration, and resistance to conventional therapy [1,2]. Current management—surgical resection followed by radiotherapy and temozolomide—yields a median survival of only ~15 months and a five-year survival rate under 10% [3]. These bleak outcomes underscore the urgent need for new, more effective therapeutic strategies.

The infiltrative growth of glioblastoma prevents complete surgical removal [4], and its pronounced inter- and intratumor heterogeneity hampers the development of uniform, targeted therapies [5]. A protective microenvironment further shields the cancer from immune attack and drug penetration, fostering recurrence [6]. Consequently, current research is pivoting to precision strategies that match treatment to each patient’s molecular profile in order to identify actionable targets [7].

Beyond its biological intractability, glioblastoma inflicts severe cognitive, psychological, and social burdens that demand holistic management—combining tumor-directed therapy with symptom control, psychosocial support, and palliative care [8]. Because the tumor routinely evades current multimodal treatment, meaningful progress will hinge on elucidating the molecular drivers of invasion and resistance and translating those insights into precisely targeted interventions.

Glioblastoma is a family of molecularly distinct entities whose behavior tracks closely with the transcriptional subtype. Among them, the proneural (PN) class is enriched for OLIG2, PDGFRA and neural lineage programs, whereas the mesenchymal (MES) class overexpresses CD44, CHI3L1 (YKL-40), and inflammatory drivers and is associated with rampant invasion, radio-chemoresistance, and the poorest survival. Growing evidence points to extracellular alarmins—particularly High-Mobility-Group-Box 1 (HMGB1)—and its cognate receptor RAGE as key cues that remodel the tumor microenvironment [9,10], and bias GBM cells toward a MES fate [11,12,13,14], yet the mechanistic link between HMGB1-RAGE signaling, epithelial-to-mesenchymal transition (EMT) circuitry, and metabolic plasticity remains incompletely resolved.

To model the proneural–mesenchymal spectrum, we compared U87-MG and GBM8401 cells. U87-MG (PN/classical) lacks EGFR amplification, carries a hemizygous PTEN loss with partially methylated MGMT, and expresses high OLIG2/PDGFRA/SOX2 yet low CD44/CHI3L1, forming compact, weakly invasive spheroids. GBM8401 is MES-like—harboring TP53^R273C^, EGFR amplification/ΔvIII, unmethylated MGMT, abundant CD44/CHI3L1/VIM, a sizeable ALDH^high^/CD44^+^/CD133^+^ stem cell pool, and three-fold greater, highly invasive sphere growth—making this pair an ideal platform for dissecting HMGB1-RAGE-driven PN-to-MES transition and testing subtype-selective therapies.

TCGA GBM data, patient IHC, and contrasting PN-type U87-MG with MES-type GBM8401 cells all show strong upregulation of HMGB1 and RAGE in the mesenchymal state. Genetic knockdown of either gene diminishes EMT markers, motility, stemness, and the dual oxidative–glycolytic metabolism that typifies MES GBM, whereas alteration in HMGB1 or RAGE expression steers PN cells toward a MES transcriptional and metabolic profile. These findings position the HMGB1-RAGE axis as a central driver of mesenchymal identity and treatment resistance and point to its blockade as a feasible means to halt PN-to-MES transition and enhance glioblastoma therapy.

## 2. Results

### 2.1. Aberrant Expression of HMGB1 and RAGE in Glioma and Its Association with Clinical Outcomes

Bioinformatic analyses show that HMGB1 and RAGE are markedly overexpressed in glioma, and their elevated levels closely track with poorer clinical outcomes. The mRNA expression level of HMGB1/RAGE in glioma clinical samples exceeds that of their nontumor variants (Figure 1). Both the TCGA-GBM and LGG datasets confirmed that glioma tumors display an induced expression of HMGB1 and RAGE, as presented in Figure 1A. These datasets, as shown in this figure, demonstrate that HMGB1 and RAGE exhibit high expression levels in GBM compared to several conditions, including oligodendroglioma, oligoastrocytoma, and astrocytoma. A statistical analysis of the Kaplan–Meier overall survival plot was performed from gene expression datasets of brain lower-grade glioma and glioblastoma multiforme (GBM/LGG) obtained from TCGA (*n* = 1152). The data presented in Figure 1B demonstrates that patient survival in brain tumors depends on the expression levels of both genes. Our TCGA-GBM/LGG datasets show a very strong positive expression correlation between HMGB1 and RAGE, with r^2^ = 0.35 and a *p*-value of 0.0172 (Figure 1C). The result of this study indicates that these genetic elements share potential functions in the progression and clinical outcomes of glioma tumors.

### 2.2. Immunohistochemical Analysis of HMGB1 and RAGE in GBM SHH In-House Clinical Samples

Immunohistochemical analysis of the Taipei Medical University–Joint Biobank cohort revealed a clear upregulation of both HMGB1 and RAGE in glioblastoma (GBM) and in intracranial secondary lesions (brain metastases of mixed primary origin) compared with normal brain tissue (Figure 2A). Quantification using the Q-score (Σ [%-positive cells × intensity grade 0–3]; range 0–300) confirmed significantly higher expression values in both primary GBM and unstratified metastatic brain lesions (Figure 2B; *p* < 0.05, *p* < 0.01, *** *p* < 0.001, one-way ANOVA). Stratification of patients at the median Q-score further demonstrated that the HMGB1/RAGE-high subgroup exhibited markedly shorter overall survival compared with the HMGB1/RAGE-low subgroup (log-rank *p* < 0.001 and *p* < 0.05; Figure 2C,D). Collectively, these findings establish HMGB1 and RAGE as frequently overexpressed in GBM and intracranial secondary lesions of mixed primary origin, highlighting HMGB1/RAGE as an adverse prognostic biomarker and a potential therapeutic target, while also noting that future stratification by the site of primary tumor origin will be important.

### 2.3. HMGB1 Knockdown Attenuates Migration and Invasion of GBM8401 Cells

Migration and invasion are key contributors to glioblastoma lethality. We found that HMGB1 depletion in GBM8401 cells significantly suppresses both processes, as shown in Figure 3. Stable knockdown of HMGB1 using two independent shRNAs (#1 and #2) led to a marked reduction in the protein levels of *N*-cadherin, vimentin, and Snail, all of which are critical regulators of the epithelial–mesenchymal transition (EMT). This downregulation is clearly shown in the Western blot results (Figure 3A), with quantification values from three independent experiments presented in red. Consistently, qRT-PCR analysis confirmed an efficient and significant inhibition of HMGB1 expression in both shHMGB1#1 and shHMGB1#2 cells, as illustrated by the bar plots (Figure 3B). Functionally, HMGB1 knockdown markedly reduced cell migration in wound-healing assays, where the closure of the scratch was substantially impaired after 24 h compared with control cells (Figure 3C, left panel), and this effect was further validated by quantitative analysis (Figure 3C, right panel). Similarly, invasive behavior was strongly diminished in Matrigel invasion assays, as evidenced by the fewer invading cells in the shHMGB1 groups relative to controls (Figure 3D, left panel), supported by corresponding quantification (Figure 3D, right panel). Data are presented as mean ± S.E.M. from three independent experiments, with statistical significance indicated as *p* < 0.05, * *p* < 0.01, and ** *p* < 0.001. Collectively, these findings demonstrate that HMGB1 promotes EMT, migration, and invasion in GBM8401 cells, underscoring its potential as a therapeutic target for glioblastoma progression.

### 2.4. ELISA Analysis of HMGB1 and RAGE Expression

Because HMGB1 is released as an alarmin and soluble RAGE amplifies its downstream signaling, their extracellular concentrations serve as indicators of pathway activity and tumor aggressiveness. ELISA confirmed that mesenchymal-type GBM8401 cells secreted higher levels of HMGB1 (~50 ng mL^−1^) and RAGE (~45 ng mL^−1^) compared with proneural U87-MG cells (~30 ng mL^−1^ and ~20 ng mL^−1^, respectively; Figure 4A and Appendix A). shRNA-mediated knockdown of HMGB1 or RAGE reduced their respective protein levels by ~66% and ~64% and also partially decreased the partner protein (Figure 4B, Appendix A). When GBM8401 and U87-MG cells were treated with temozolomide (200 µM, 72 h), secreted HMGB1 and RAGE levels were further reduced to ~17 ng mL^−1^ and ~23 ng mL^−1^, respectively (*p* < 0.05; Figure 4C). However, because this TMZ treatment condition is cytotoxic and substantially reduces the cell number, the decreased HMGB1 and RAGE concentrations may partly reflect lower cell density rather than a direct effect on this signaling axis. Accordingly, these findings should be interpreted with caution and considered supportive rather than conclusive evidence of HMGB1–RAGE modulation by TMZ.

### 2.5. Comparative Metabolic Analysis of Proneural and Mesenchymal Glioblastoma Cells

The mesenchymal transition in glioblastoma is frequently accompanied by metabolic reprogramming, which enhances invasion and therapy resistance. To investigate this, we compared the bioenergetic profiles of proneural (U87-MG) and mesenchymal-like (GBM8401) cells using Seahorse XF analysis. Mesenchymal GBM8401 cells exhibited a significantly higher basal oxygen consumption rate (OCR; +30%, *p* < 0.001) and extracellular acidification rate (ECAR; +50%, *p* < 0.001) compared with proneural U87-MG cells, and these differences were maintained throughout sequential injections of oligomycin, FCCP, and rotenone/antimycin A (Figure 5A–D). Quantitative analysis revealed that ATP-linked respiration, spare respiratory capacity, and glycolytic reserve were each elevated by approximately 30–40% in GBM8401 cells (*p* < 0.01), consistent with a dual-fuel phenotype relying on both mitochondrial respiration and glycolysis. These findings underscore that mesenchymal-like glioblastoma cells possess enhanced metabolic flexibility, which may contribute to their invasive and drug-resistant behavior and highlight potential therapeutic vulnerabilities in targeting both oxidative phosphorylation and glycolytic pathways.

### 2.6. Silencing RAGE Expression Suppresses the Growth Potential of Glioma Stem Cells In Vitro

To assess whether RAGE contributes to the stem-like phenotype of glioma cells, we performed a 3D tumorsphere assay—a recognized surrogate for self-renewal (Figure 6A). Immunoblot analysis confirmed that RAGE expression is associated with elevated levels of pluripotency factors such as Oct4 in sphere-forming GBM8401 cultures (Figure 6B). Conversely, shRNA-mediated silencing of RAGE reduced both the number and average size of spheres (Figure 6C), indicating that RAGE is required to sustain glioma stem cell capacity. In support of this, STRING interaction analysis placed the HMGB1–RAGE axis at the center of a network connecting EMT and CSC markers (Figure 6D), implicating this pathway as a driver of tumor plasticity and aggressiveness.

### 2.7. Expression of HMGB1 Associates with Mesenchymal and Proneural Markers in Glioblastoma Behavior and Clinical Outcome

Because mesenchymal transition and stem cell plasticity underlie glioblastoma aggressiveness, we next examined whether HMGB1 actively contributes to the PN-to-MES switch in tumor samples, patient-derived glioblastoma organoids (GBOs), and stem-like cells. RNA-seq of GBM8401 cells transduced with shHMGB1 revealed broad suppression of canonical MES genes (CD44, CHI3L1, VIM, FN1) and concurrent upregulation of PN genes (OLIG2, PDGFRA, SOX10) (Figure 7A,B). In clinical tissues, immunohistochemistry confirmed abundant CD44 (MES marker) and reduced CD133 (PN marker) expression, with HMGB1 preferentially localized to CD44^+^ tumor regions (Figure 7C,D). Consistent with these patterns, HMGB1 knockdown shifted a panel of glioma stem cell (GSC) markers from a MES-GSC profile (WT1, LYN, CD44, SLC2A1) toward a PN-GSC profile (CD133, SOX2, NOTCH1) (Figure 7E). qRT-PCR validation supported these RNA-seq findings (Figure 7F). Together, these results implicate HMGB1 as a molecular driver of the mesenchymal, CD44^+^/CD133^−^ stem-like phenotype, reinforcing its role as a therapeutic target in glioblastoma.

## 3. Discussion

In this study, we examined the role of High Mobility Group Box 1 (HMGB1) and its receptor RAGE in glioblastoma (GBM) progression. Analysis of TCGA-GBM, LGG datasets, and in-house SHH-hospital patient data revealed elevated HMGB1 and RAGE expression in glioma tissues, correlating with poor survival and accelerated tumor growth, consistent with previous reports [2,15,16,17]. Evidence in the literature supports the conclusion that high HMGB1 levels are associated with tumor aggressiveness and poor prognosis [1,18], aligning with our patient cohort findings. Prior studies also indicate that HMGB1 activates ERK and NF-κB signaling, promoting proliferation and invasion [19]. Our experimental data confirmed that HMGB1 knockdown reduces cell migration, invasion, EMT, and CSC markers. Moreover, HMGB1 has been shown to induce mesenchymal transformation and enhance invasiveness, contributing to GBM aggressiveness and therapy resistance [20,21]. Collectively, these findings underscore the prognostic significance of HMGB1 and RAGE, as confirmed by immunohistochemistry, and highlight their potential as therapeutic targets.

The presence of glioblastoma stem cells (GSCs) in GBM contributes significantly to its aggressive nature and resistance to therapy [22]. Here, we demonstrate that concurrent suppression of RAGE and its ligand HMGB1 impairs critical GSC properties, including tumor sphere formation and the expression of key transcription factors involved in self-renewal in GBM8401 cells [23,24]. These findings align with previous studies reporting that elevated HMGB1 expression and heightened RAGE activity are associated with poor prognosis in glioblastoma patients, as both factors promote tumor progression and serve as prognostic indicators [25,26]. Consistently, RAGE signaling has been implicated in the maintenance of GSC phenotypes, contributing to therapy resistance and tumor recurrence [11,12,27,28]. Further studies are required to delineate the molecular mechanisms by which HMGB1-RAGE axis regulates self-renewal capacity in GSCs.

Hypoxia represents a major microenvironmental stressor that induces HMGB1 release, thereby activating RAGE-dependent signaling cascades and promoting glioblastoma progression [29,30,31]. In agreement with prior studies, we observed that hypoxic tumor regions exhibit elevated HMGB1 and RAGE expression, emphasizing the contribution of microenvironmental stress to mesenchymal reprogramming and aggressive phenotypes. Secreted HMGB1–RAGE complexes likely function as mediators of tumor–stroma interactions, amplifying pro-inflammatory signaling and angiogenesis to accelerate tumor growth and immune evasion [2,32,33]. Furthermore, the attenuated migratory and invasive behavior of HMGB1-deficient cells observed in our study is consistent with NFκB-dependent mechanisms previously implicated in HMGB1–RAGE-driven proliferation, invasion, and metastasis [34,35].

To clarify subtype-specific vulnerabilities, we compared proneural U87-MG with mesenchymal GBM8401 cells. Echoing earlier reports that HMGB1 and RAGE are enriched in MES tumors and linked to resistance [36,37], GBM8401 secreted almost twice the HMGB1 and RAGE of U87-MG (≈39 vs. 30 ng mL^−1^ and 33 vs. 20 ng mL^−1^, respectively; ELISA). Silencing either gene collapsed the axis in both directions (shHMGB1: HMGB1 39 → 5.8, RAGE 33 → 11.5 ng mL^−1^; shRAGE: RAGE 33 → 5.6, HMGB1 39 → 11.1 ng mL^−1^), revealing a positive feedback loop that locks cells into the MES state. Temozolomide produced a similar, though partial, suppression (HMGB1 40 → 17; RAGE 43.5 → 23.4 ng mL^−1^; *p* < 0.05) accompanied by fewer spheres and more apoptosis, consistent with alarmin-driven drug tolerance [38]. Thus, HMGB1-RAGE is not merely a MES marker but an active driver whose disruption—genetically or pharmacologically—attenuates the invasive, stem-like, metabolically flexible phenotype of mesenchymal glioblastoma.

These findings position the HMGB1–RAGE axis as an actionable therapeutic node whose modulation could augment current chemotherapeutic regimens and improve outcomes in mesenchymal-rich glioblastoma [11]. The reduction in HMGB1 and RAGE levels observed in this study aligns with earlier findings, where the inhibition of HMGB1 and RAGE signaling attenuated glioblastoma cell migration and invasion, reducing tumor aggressiveness [34]. Additionally, the observed reduction in HMGB1 and RAGE protein levels following temozolomide treatment aligns with research that has shown that targeting the HMGB1-RAGE pathway can enhance the sensitivity of glioblastoma cells to chemotherapy, specifically in the MES subtype [35].

Large-scale metabolic reprogramming, marked by elevated mitochondrial respiration and glycolysis, is a defining feature of aggressive tumors, including MES GBM [39]. While these pathways offer promising therapeutic targets [36], metabolic redundancies often undermine their clinical exploitability. Our OCR and ECAR analyses revealed a pronounced increase in both oxidative phosphorylation and glycolytic flux in MES GBM cells, highlighting key metabolic adaptations during subtype transition. These findings are consistent with recent studies [11] identifying the HMGB1-RAGE signaling axis as a central driver of metabolic reprogramming and stemness in GBM. We also observed activation of epithelial-to-mesenchymal transition (EMT), evidenced by upregulation of *N*-cadherin and Snail, supporting prior reports that HMGB1-induced p62 expression promotes Snail-mediated EMT and invasion [40]. Furthermore, the enhanced tumor sphere formation and stemness marker expression observed in our study parallel findings that HMGB1 released by autophagic cancer-associated fibroblasts maintains cancer cell stemness via RAGE in breast cancer models [41]. Collectively, these results position the HMGB1-RAGE axis as a pivotal regulator of metabolic adaptation, EMT, and stemness in GBM. Whether this pathway can be effectively targeted for therapeutic benefit remains an important and promising area for future investigation. A limitation of this study is that the brain metastases analyzed were of mixed, unstratified primary origin. As metastatic biology varies by primary tumor site, our findings should be interpreted as representative of brain metastases in general rather than site-specific subgroups.

Our results place the HMGB1-RAGE axis at the center of glioblastoma plasticity: HMGB1 binding to RAGE activates NF-κB/STAT3, drives *N*-cadherin/Snail-mediated EMT, and boosts Oct4-linked tumorsphere formation, thereby coupling mesenchymal identity to stemness and metabolic flexibility—findings that echo earlier work in GBM and breast cancer models [40,41]. Blocking either partner, genetically or with small-molecule antagonists such as glycyrrhizin or the BBB-penetrant RAGE inhibitor FPS-ZM1, curbs migration, invasion, and bioenergetic reprogramming in pre-clinical studies [3,7,10,42,43], highlighting this pathway as a tractable target to prevent the proneural-to-mesenchymal switch and enhance temozolomide efficacy. Figure 8 summarizes the model: extracellular HMGB1 engages RAGE to initiate EMT and stemness, whereas RAGE blockade severs the loop, attenuating invasion and therapy resistance.

## 4. Materials and Methods

### 4.1. Study Design and Ethical Approval

This study was designed to investigate the role of HMGB1–RAGE signaling in glioblastoma progression using a combination of bioinformatics analyses, patient-derived tissue samples, and in vitro functional assays. A total of 70 glioblastoma patients provided tumor tissue specimens through the Taipei Medical University–Joint Biobank. Both primary and recurrent GBM samples were included. For each case, tumor tissues were processed in two ways: (i) a portion of freshly excised tissue was snap-frozen and reserved for molecular studies, including RNA sequencing and qRT-PCR, and (ii) matched samples were formalin-fixed and paraffin-embedded (FFPE) for histopathological evaluation and immunohistochemistry. All procedures were approved by the Joint Institutional Review Board (JIRB) of Taipei Medical University–Shuang Ho Hospital (Approval no.: N202101069, NBCT No.200039, NBCT No.220104, N202101065, N201801070, N202002052, and N202008022). The use of patient tissues complied with the Declaration of Helsinki for Biomedical Research, and informed consent was obtained from all participants or their legal representatives.

### 4.2. Sample Collection and Patient Data Analysis

Glioblastoma tissue samples collected and stored in the Taipei Medical University–Joint Biobank were selected based on available clinical data, including tumor grade, overall survival, and treatment response. Fresh-frozen tissues were used for RNA extraction and sequencing, followed by bioinformatics analysis, and expression of selected genes was validated by qRT-PCR. Matched FFPE samples were used for histopathological assessment, including immunohistochemistry. For subtype comparisons, histopathological and molecular profiling was used to distinguish mesenchymal (MES) and proneural (PN) glioblastoma subtypes.

### 4.3. Gene Expression Profiling

This study analyzed expression data of HMGB1 and RAGE from samples obtained from The Cancer Genome Atlas (TCGA-GBM, LGG). The utility of Gene Expression Profiling Interactive Analysis (GEPIA) assessed the connection between HMGB1 and RAGE expression concerning patient prognosis. Data from RNA sequencing helped evaluate the correlation effect on patient survival statistics for these genes with statistical significance.

### 4.4. Survival Analysis

Survival analysis using the Kaplan–Meier method evaluated the overall survival (OS) according to HMGB1 and RAGE expression levels in patients with glioblastoma. Procedure evaluation used the Cox regression model to determine survival effects in patients while accounting for potential confounders. *p* values less than 0.05 determined statistical significance through a log-rank test.

### 4.5. Immunohistochemistry (IHC) and Tissue Microarrays

Glioblastoma tissue samples were formalin-fixed and paraffin-embedded (FFPE) and then sectioned on glass slides to a 5 µm thickness. The first step used xylene as a deparaffination agent, followed by ethanol solutions to rehydrate tissue sections. The procedure for epitope retrieval started with treatment in citrate buffer (pH 6.0) at 95 °C for 20 min. Per the protocol, the primary antibodies HMGB1 (Abcam ab79823, Waltham, MA, USA) and RAGE (Cell Signaling Technology 21192, Danvers, MA, USA) were incubated at 4 °C overnight. The detection process included HRP-conjugated goat anti-rabbit IgG as the secondary antibody, followed by a DAB (diaminobenzidine) reaction solution. The staining intensity and distribution were quantified using ImageJ software (National Institutes of Health, Bethesda, MD, USA; available at https://imagej.net/ij/, accessed 8 May 2025). To assign numerical values to staining intensity, Q score analysis was used, with a score that falls between 0 and 300. Student’s *t*-test was applied to the staining intensity between the MES and PN glioblastoma subtypes for statistical analysis. Statistical significance between groups was considered at *p* < 0.05.

### 4.6. Cell Culture

Human glioblastoma lines GBM8401 and U87-MG (all purchased from the American Type Culture Collection, ATCC, Manassas, VA, USA) were used. U87-MG (HTB-14) represents a proneural/classical phenotype characterized by high OLIG2/PDGFRA and low CD44/CHI3L1, whereas GBM8401 (CRL-5989) is mesenchymal-like, harboring TP53R273C and amplified/ΔEGFRvIII with strong CD44/CHI3L1 expression. All lines were cultured at 37 °C, 5% CO_2_, in DMEM supplemented with 10% heat-inactivated fetal-bovine serum, 100 U mL^−1^ penicillin, and 100 µg mL^−1^ streptomycin. Cells were passaged at 80–90% confluence and experiments were performed with passages 4–12 to minimize phenotypic drift.

### 4.7. shRNA-Mediated Gene Knockdown

To minimize off-target effects, we designed three candidate RNA interference motifs against HMGB1 and two against RAGE and then selected the most potent sequences for functional studies. The HMGB1 hairpin ultimately used carried the sense motif 5′-CACCGCGAAGAAACTGGGAGAGATG-3′ (full oligo: 5′-CACCGCGAAGAAACTGGGAGAGATGTTCAAGAGACATCTCTCCCAGTTTCTTCGCTTTTTTG-3′). For RAGE, two non-overlapping 21-nt targets were synthesized: 5′-GAACTGAATCAGTCGGAGGAA-3′ and 5′-GTGGAGATCTTGTAAACTTAA-3′. A scrambled control with no homology to human transcripts (5′-CCTAAGGTTAAGTCGCCCTCGCTC-3′) served as the negative control. All oligonucleotides were annealed and ligated into the Bbs I site of the pGPU6/GFP/Neo vector (GenePharma, Taipei, Taiwan). Insert orientation and fidelity were confirmed by Sanger sequencing. GBM8401 and U87-MG cells were seeded at 5 × 10^4^ cells per well in six-well plates 24 h before transfection to reach ~70% confluence. Transfection complexes were prepared in 250 µL Opti-MEM I (Invitrogen, Carlsbad, CA, USA) containing 7.5 µL plasmid DNA and 5 µL Lipofectamine 2000 (Invitrogen) per well, incubated for 20 min at room temperature, and then added dropwise. Plates were returned to 37 °C/5% CO_2_ for 48–72 h; GFP fluorescence routinely exceeded 80%, indicating high transfection efficiency. For stable pools, cells were selected with 500 µg ml^−1^ G418 (Gibco, Cat. 10131035) for 7–10 days and expanded for experiments at passages 4–10.

### 4.8. ELISA

ELISA was employed to quantify secreted HMGB1 and RAGE. Culture supernatants were collected from mesenchymal-like and proneural-like glioblastoma monolayers 48 h after plating (or 48 h after shRNA transduction where indicated). The medium (1.5 mL) was aspirated from each six-well dish, cleared of debris by 5 min centrifugation at 1500× *g*, and stored on ice. Aliquots (100 µL) were analyzed in duplicate with commercially available human HMGB1 and human RAGE DuoSet ELISA kits (R&D Systems, Minneapolis, MN, USA) according to the manufacturer’s instructions. Briefly, standards (0–2000 pg mL^−1^) and samples were incubated for 2 h in antibody-coated 96-well plates, followed by biotinylated detection antibody, streptavidin–HRP, and TMB substrate; the reaction was terminated with 2 N H_2_SO_4_, and absorbance was read at 450 nm (reference 570 nm) on a BioTek Synergy H1 plate reader. Protein concentrations (pg mL^−1^) were interpolated from four-parameter logistic standard curves and normalized to the number of viable cells present at collection (trypan-blue exclusion). Three biological replicates were analyzed per condition. Data are presented as mean ± SD; statistical significance was evaluated with two-tailed Student’s *t*-tests, accepting *p* < 0.05 as significant.

### 4.9. Invasion and Cell Migration Assays

Polycarbonate inserts (8 µm pore, 6-well format; Corning, Corning, NY, USA) served as Boyden chambers. For invasion assays, each insert was coated with 50 µL ice-cold growth-factor-reduced Matrigel (1 mg mL^−1^) and then polymerized for 30 min at 37 °C; uncoated inserts were used for the migration arm. Exponentially growing cells were harvested, washed, and resuspended in serum-free RPMI containing 2% bovine serum albumin. We seeded 1 × 10^5^ U87-MG or GBM8401 cells in 500 µL into the upper compartment, while 2.5 mL RPMI with 20% FBS was added below as a chemoattractant. Drug or vehicle treatments, when applied, were included in both chambers. After 48 h at 37 °C/5% CO_2_, cells remaining on the upper membrane surface were gently removed with cotton swabs. Inserts were fixed in 70% ethanol for 10 min and stained with 0.5% crystal violet. Five randomly chosen 200× fields per insert were imaged, and crystal-violet-positive cells on the underside of the membrane were counted by two investigators blinded to the experimental group, yielding quantitative indices of migration and invasion for each cell line.

### 4.10. Wound-Healing Migration Assay

Two-dimensional cell migration was assessed using culture inserts designed for wound-healing assays (ibidi GmbH, Munich, Germany). Glioblastoma cells were seeded into both chambers of the insert and maintained in complete medium at 37 °C in a humidified atmosphere containing 5% CO_2_ until confluence. The silicone barrier separating the chambers was then carefully removed to create a uniform cell-free gap. Monolayers were gently washed with PBS (Gibco, Waltham, MA, USA) to remove detached cells and subsequently cultured in serum-reduced medium (2% FBS; Gibco, Grand Island, NY, USA; Cat. #16000044, Lot #2297076) to minimize proliferation during the assay. Wound closure was monitored for 24 h, with images acquired every 2 h using a BioTek Lionheart FX automated imaging system (Agilent Technologies, Santa Clara, CA, USA) equipped with a phase-contrast microscope. The percentage of wound closure was quantified as a measure of cell migratory capacity.

### 4.11. Tumor Sphere Formation Assay

Single-cell suspensions were prepared with Accutase, passed through a 40 µm strainer, and seeded at 1 × 10^4^ cells per well in 2 mL of serum-free DMEM/Ham’s F-12 (1: 1) into six-well ultra-low-attachment plates (Corning). The sphere medium contained human EGF (20 ng mL^−1^), basic FGF (20 ng mL^−1^; PeproTech, Rocky Hill, NJ, USA), heparin (2 µg mL^−1^; Sigma, St. Louis, MI, USA), and 1% penicillin/streptomycin. Cultures were maintained at 37 °C, 5% CO_2_; half of the medium (1 mL) was gently replenished every three days. After 14 days, each well was imaged with an inverted phase-contrast microscope (4 × objective); nine non-overlapping fields were captured and analyzed in ImageJ. Compact, non-adherent aggregates with an equivalent diameter ≥ 50 µm were scored as tumor spheres. Sphere-forming efficiency (SFE) was calculated as
SFE (%) = (number of spheres (≥50 µm) ÷ 1 × 10^4^ seeded cells) × 100

Two blind observers independently counted each field; values differed by <5%, and the mean was used for statistics.

### 4.12. Western Blot Analysis

After the experiment, the treated cells were collected and proteins were isolated with RIPA buffer (Sigma), which included protease inhibitors and phosphatase inhibitors from Sigma-Aldrich and boiled the proteins for denaturation. The BCA Protein Assay Kit (Pierce, Appleton, WI, USA) provided a method for determining extract protein concentration. The SDS-PAGE gel received 50 µg of protein material before PVDF membrane transfer took place (Bio-Rad, Hercules, CA, USA). The analysis began with blocking the membranes by adding 5% non-fat dry milk, after which the researchers applied primary antibodies, including HMGB1, RAGE, *N* cadherin, Snail, and β actin. The band was detected with ECL (Thermon Fisher, Waltham, MA USA) after exposing it to HRP-conjugated secondary antibodies. ImageJ software analyzed the protein data after normalization to β-actin based on the densitometric analysis.

### 4.13. Seahorse Extracellular Flux Assay

Measurement of mitochondrial respiration and glycolytic metabolism occurred through analysis with Seahorse XF24 Extracellular Flux Analyzer (Agilent Technologies, Santa Clara, CA, USA). Cells received a seeding density of 20,000 cells/well distributed in Seahorse XF24 plates (Santa Clara, CA, USA) before overnight incubation. Before running the assay, we added XF assay media to the culture media while cells reached equilibrium at 37 °C in a non-CO_2_ incubator for one hour. The measurement of baseline OCR and ECAR took place after adding metabolic inhibitor sequences of oligomycin (1 μM) and FCCP (1 μM) and rotenone/antimycin A (both 2 μM) to the cell culture and data were analyzed using Seahorse Wave software (version 2.6.1, Agilent Technologies; available at https://www.agilent.com/en/product/cell-analysis/real-time-cell-metabolic-analysis/xf-software/seahorse-wave-desktop-software-740897, accessed on 18 May 2025). 

### 4.14. qRT-PCR

TRIzol reagent (Thermo Fisher, Waltham, MA, USA) extracted total RNA using the iScript cDNA Synthesis Kit (Bio-Rad). Subsequently, qRT-PCR was conducted using SYBR Green Master Mix (Bio-Rad) about the internal control β-actin. Scientists measured the relative expression levels of HMGB1, RAGE, *N*-cadherin Snail and other genes through the 2^−ΔΔCT^ analysis technique.

### 4.15. Statistical Analysis

Statistical analysis was conducted through GraphPad Prism 7. SEM values surround each mean value in the presented results. Two-group comparisons used Student's *t* test, and one-way ANOVA with Tukey’s multiple comparison test served to assess differences between three or more groups. Survival rates of glioblastoma patients were analyzed through Kaplan–Meier survival curves, together with log-rank test evaluation for statistical significance. The study considered statistical significance as any *p*-value smaller than 0.05.

## 5. Conclusions

The HMGB1–RAGE axis contributes to glioblastoma aggressiveness by promoting EMT, stemness, and metabolic flexibility, which reinforce mesenchymal traits. Knockdown of either partner impairs migration, invasion, and tumorsphere formation, underscoring their role in tumor plasticity. While our analyses support HMGB1–RAGE as a therapeutic target, we acknowledge limitations, including the absence of Seahorse profiling in RAGE-silenced cells, lack of in vivo validation, and the need for caution in inferring full PN-to-MES conversion. Future studies addressing these gaps will be essential to confirm the translational potential of targeting this pathway in GBM.

## Figures and Tables

**Figure 1 ijms-26-09352-f001:**
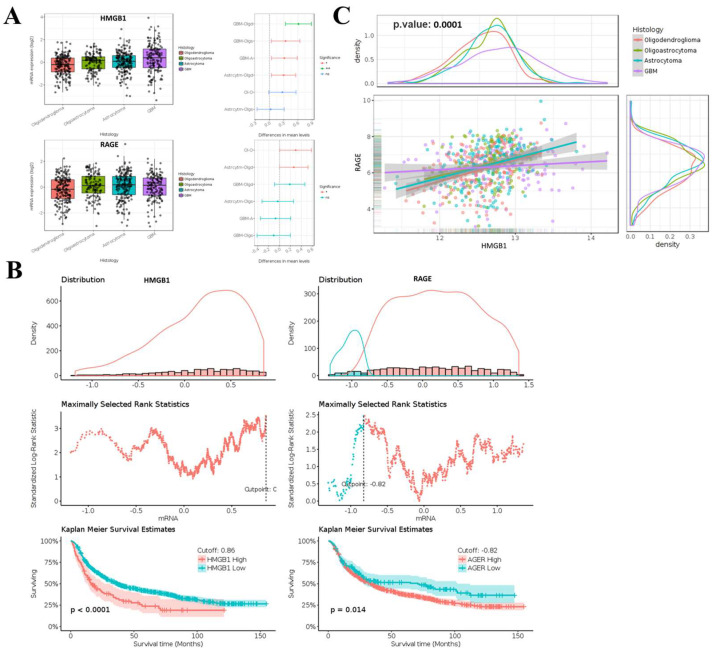
**Expression analysis of HMGB1 and RAGE in TCGA_GBM-LGG datasets.** (**A**) Induced expression of mRNA levels of HMGB1 and RAGE in TCGA-GBM and LGG datasets. Boxplots show median ± IQR with individual data points. Statistical significance was determined by one-way ANOVA with Tukey’s post hoc test: * *p* < 0.05, *** *p* < 0.001, ns = not significant. (**B**) Kaplan–Meier survival analysis plot of HMGB1 and RAGE expression in TCGA-GBM and LGG gene expression database by RNAseq (polyA+ IlluminaHiSeq). (**C**) TCGA-GBM/LGG data reveal that HMGB1/RAGE expression has a strong and positive correlation, as shown in r^2^ value of 0.35 at *p*-value 0.001.

**Figure 2 ijms-26-09352-f002:**
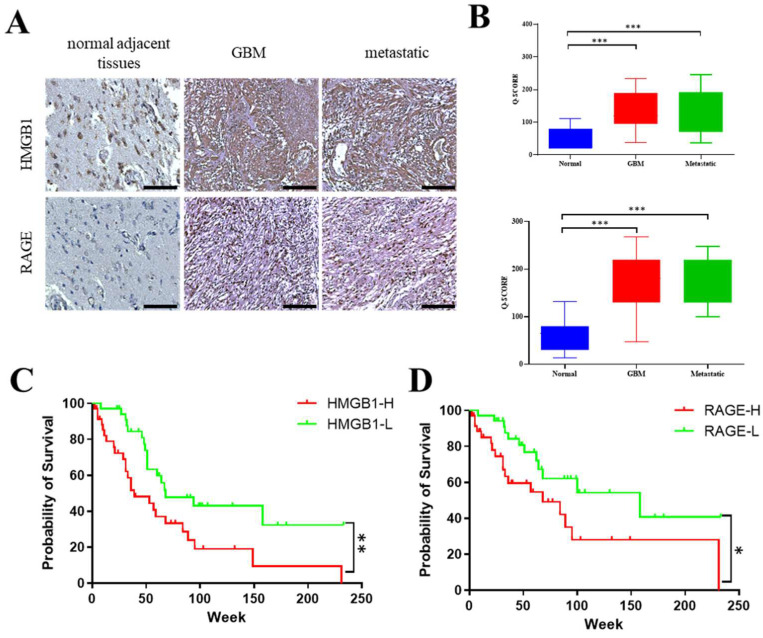
**Elevated HMGB1/RAGE expression in glioblastoma correlates with reduced overall survival.** (**A**) Representative immunohistochemistry (IHC) staining of HMGB1 and RAGE in normal brain parenchyma, primary glioblastoma (GBM), and intracranial secondary lesions (brain metastases of mixed primary origin). Scale bar = 100 µm. (**B**) Semi-quantitative evaluation of IHC signals using the Q-score (Σ [% positive cells × staining intensity grade 0–3]; range 0–300). Both GBM and unstratified brain metastases displayed significantly higher Q-scores compared with normal brain tissue (*** *p* < 0.001; one-way ANOVA with Tukey’s post hoc test). (**C**,**D**) Kaplan–Meier survival curves stratified by HMGB1 and RAGE expression (HMGB1-H/RAGE-H, high; HMGB1-L/RAGE-L, low; cutoff = median Q-score) demonstrate that high HMGB1 and RAGE expression are significantly associated with shorter overall survival (** *p* < 0.01 for HMGB1, * *p* < 0.05 for RAGE; log-rank test).

**Figure 3 ijms-26-09352-f003:**
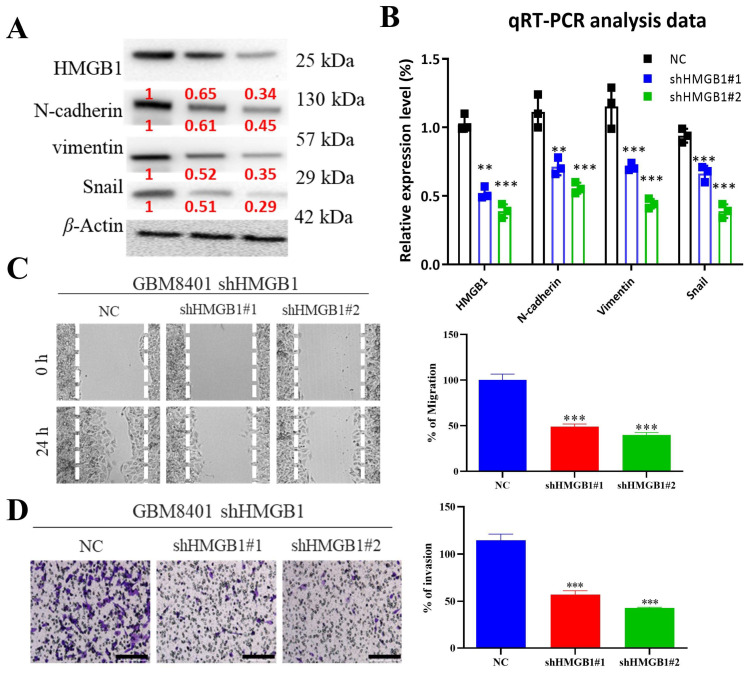
**HMGB1 knockdown inhibits migration and invasion in GBM8401 cells.** (**A**,**B**) HMGB1 shRNA (by both shRNA #1 and #2) knockdown on GBM8401 cells shows downregulation of *N*-cadherin, vimentin, and Snail, as shown in Western blot analysis and qRT-PCR analysis. Each data point averages the mean value with its standard error measurement (S.E.M.) from three separate experimental sets. (**C**) An evaluation of cellular wound-healing patterns following HMGB1 knockdown; quantification bar plot is shown on the right. (**D**) Analysis of Matrigel invasion and quantification of Matrigel invasion assay; bar plot, right side. ** *p* < 0.01, *** *p* < 0.001.

**Figure 4 ijms-26-09352-f004:**
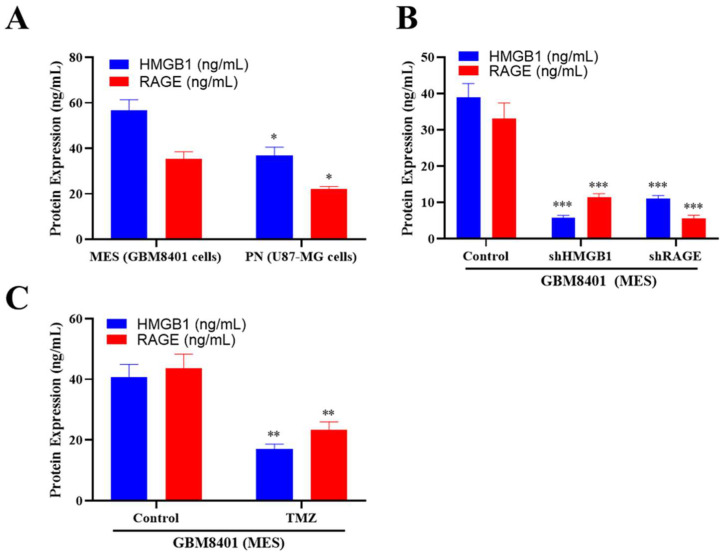
**ELISA analysis of extracellular HMGB1 and RAGE secretion in GBM cells.** (**A**) Baseline secretion in GBM8401 and U87-MG cells (*p* < 0.05). (**B**) Knockdown of HMGB1 or RAGE using shRNA reduced protein secretion in MES GBM cells (*p* < 0.05). (**C**) Temozolomide treatment (200 μM, 72 h) further reduced the secreted levels of HMGB1 and RAGE in GBM8401 cells (*p* < 0.05). The reduction observed under TMZ treatment may also reflect decreased cell density/viability at this dose and duration; therefore, these data should be interpreted with caution. * *p* < 0.05, ** *p* < 0.01, and *** *p* <0.001.

**Figure 5 ijms-26-09352-f005:**
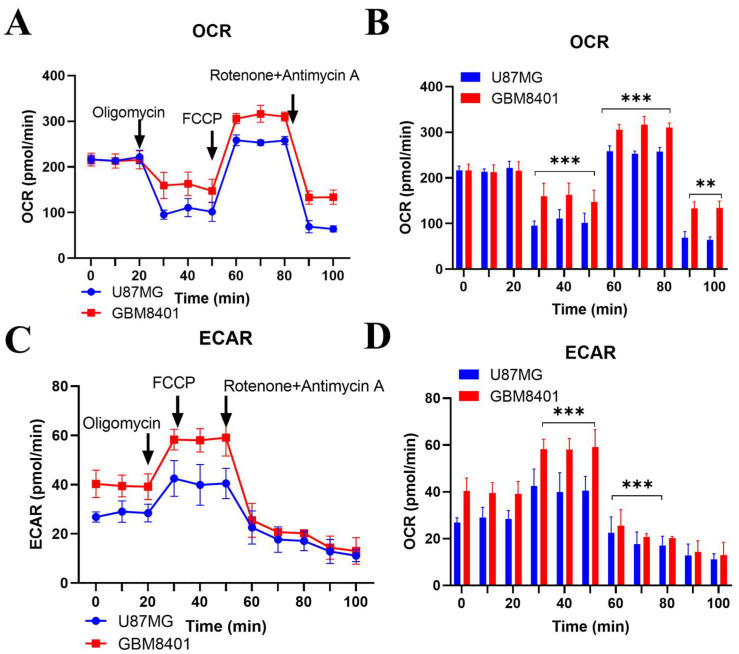
Seahorse XF-based comparison of mitochondrial and glycolytic activity in proneural (U87-MG) and mesenchymal-like (GBM8401) glioblastoma cells. (**A**) Real-time oxygen consumption rate (OCR) profile. Arrows indicate sequential injections of oligomycin (Olig; ATP synthase inhibitor), FCCP (uncoupler), and rotenone + antimycin A (Rot/AA; complex I/III inhibitors). GBM8401 cells displayed higher basal OCR, a larger oligomycin-sensitive drop (ATP-linked respiration), and a greater FCCP-stimulated peak (maximal respiration). (**B**) Quantification of OCR shown as bar plots. (**C**) Extracellular acidification rate (ECAR) from the same wells illustrates basal glycolysis, the oligomycin-induced increase defining glycolytic capacity, and the subsequent decline after FCCP. (**D**) Quantification of ECAR is shown as bar plots. Data represent mean ± SD of three independent experiments (*n* = 3). Statistical significance was assessed by two-way ANOVA followed by Tukey’s post hoc test (** *p* < 0.01, *** *p* < 0.001, vs. U87-MG).

**Figure 6 ijms-26-09352-f006:**
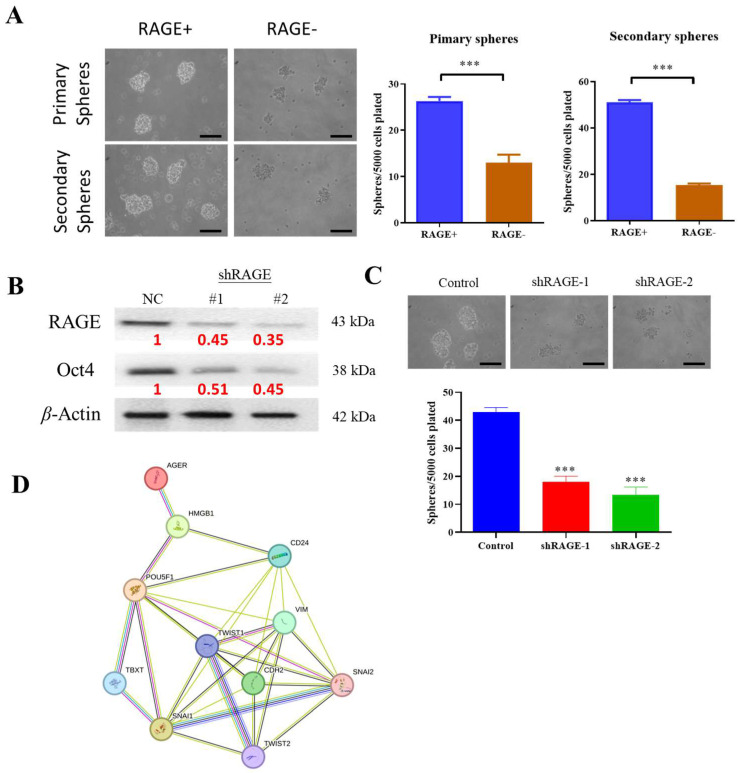
**Impact of RAGE silencing on stemness in GBM8401 cells.** (**A**) Representative images of tumor sphere formation. (*** *p* < 0.001, one-way ANOVA with Tukey’s post hoc test). Scale bar = 100 µm. (**B**) Immunoblot showing reduced expression of RAGE and pluripotency transcription factors, including Oct4, following RAGE silencing. (**C**) Quantification of tumorsphere number and size demonstrates impaired self-renewal in shRAGE-modified GBM8401 cells compared to controls. (*** *p* < 0.001, one-way ANOVA with Tukey’s post hoc test). (**D**) STRING database analysis (https://string-db.org/ (accessed on 5 May 2025)) illustrating protein–protein interaction networks linking HMGB1/RAGE with EMT and CSC markers.

**Figure 7 ijms-26-09352-f007:**
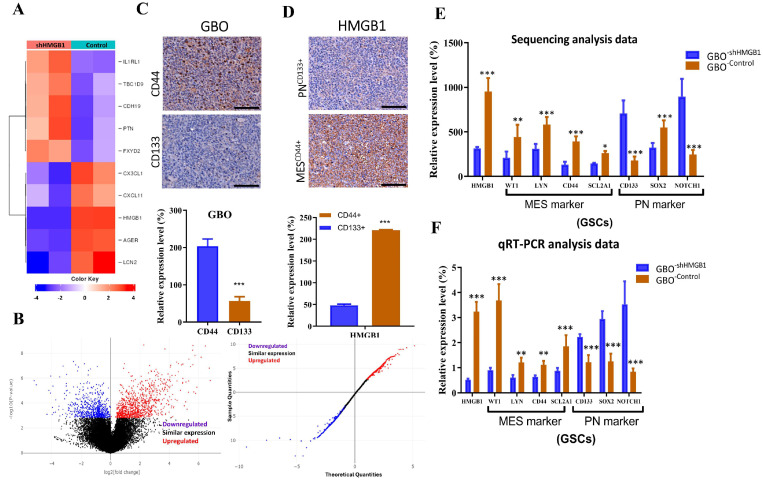
**HMGB1 bias toward the mesenchymal program in glioma stem-like cells and its association with patient outcome.** (**A**) RNA-seq heatmap (z-scored log_2_ counts) of GBM8401 cells transduced with shHMGB1 versus scramble control (*n* = 2). HMGB1 silencing suppresses MES genes and upregulates PN genes. (**B**) Volcano plot (|log_2_FC| ≥ 1, adj. *p* < 0.05). (**C**) Immunohistochemistry for CD44 (MES) and CD133 (PN) in primary GBM tissues. Bar plot quantification shows significantly higher CD44 and lower CD133 in GBM (*** *p* < 0.001; Student’s *t*-test). Scale bar = 100 µm. (**D**) HMGB1 expression enriched in MES–CD44^+^ regions and low in PN–CD133^+^ regions; bar plot shows mean ± SD Q-scores from five matched fields (*** *p* < 0.001; one-way ANOVA). (**E**) RNA-seq log_2_FC values for PN-GSC (CD133, SOX2, NOTCH1) and MES-GSC (WT1, LYN, CD44, SLC2A1) after shHMGB1.Statistical significance is indicated as * *p* < 0.05, ** *p* < 0.01, *** *p* < 0.001 versus scramble control (adjusted p-values). (**F**) qRT-PCR validation of the same panel of genes (mean ± SD, *n* = 3). Statistical significance: *** *p* < 0.001, and ** *p* < 0.01 versus scramble).

**Figure 8 ijms-26-09352-f008:**
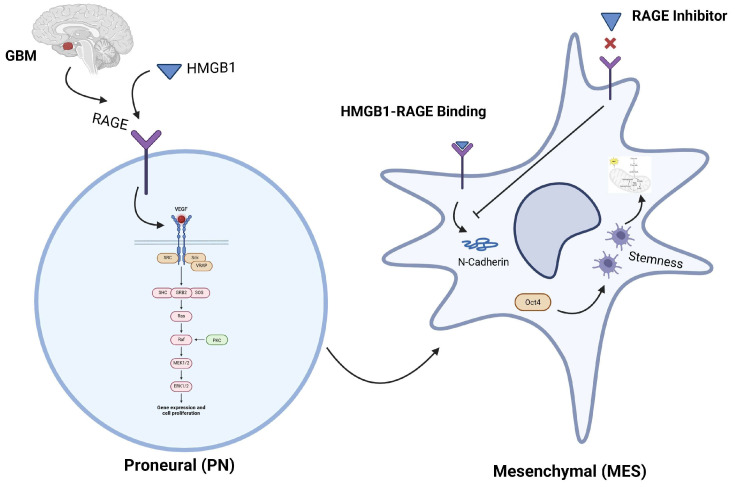
HMGB1 binds to RAGE, driving the transition from the proneural (PN) to the mesenchymal (MES) state in glioblastoma (GBM), marked by increased *N*-cadherin and OCT4 expression, promoting stemness. Blocking RAGE inhibits this shift, highlighting its potential as a therapeutic target.

## Data Availability

All experimental protocols, compound characterization data, and additional supporting information are provided in the Appendix A. Specifically, Appendix A presents CCLE transcriptomic comparison of proneural-leaning U87-MG versus mesenchymal-like GBM8401 glioblastoma cell lines. Appendix A presents HMGB1 knockdown efficiency and quantification. Appendix A presents RAGE knockdown efficiency and quantification. Appendix A includes full-size blots corresponding to Figure 3C, with results from three independent replicates. Similarly, Appendix A presents full-size blots for Figure 6B, also with three replicates. All authors have reviewed and approved the final version of the manuscript and collectively take full responsibility for the accuracy, integrity, and reproducibility of the work presented. The data used to obtain the results of this study are available from the authors upon reasonable request. The datasets used and analyzed in the current study are available from the corresponding author in response to reasonable requests.

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
