# Peer review of "The HMGB1-RAGE Axis Drives the Proneural-to-Mesenchymal Transition and Aggressiveness in Glioblastoma"

_ijms, 2025, doi:10.3390/ijms26199352_

Round 1

Reviewer 1 Report (New Reviewer)

Comments and Suggestions for Authors

Thank you for the opportunity to review this manuscript, addresses the role of HMGB1 and RAGE in glioblastoma (GBM) progression and malignancy using paraffin-embedded tissues, GBM cell lines, patient-derived cells, and bioinformatics tools. Although the study is of scientific interest, the main text—particularly the Results section—is difficult to follow due to significant grammar and language issues. A thorough revision of grammar and syntax throughout the manuscript is strongly recommended. The general conclusions are not well supported due to poor integration of results. The manuscript in its current form suffers from major issues in both scientific design and presentation, which significantly limit its impact and clarity.

Please find below my detailed comments:

  1. Please revise and update the WHO classification according to the latest CNS tumor guidelines.
  2. In Figure 1, RAGE is referred to as AGER. Please clarify and ensure consistent terminology.
  3. In line 273, the word “metastasis” appears. Please specify the tissue origin of the metastasis mentioned.
  4. The authors state: “Elevated HMGB1/RAGE expression in glioblastoma correlates with reduced overall survival,” but the Kaplan–Meier curve for RAGE is missing. Please include this data.
  5. Section 3.3 of the Results is confusing and should be rewritten for clarity.
  6. In the Western blot analysis, N-cadherin expression in Sh#1 appears higher than in the negative control (NC), which contradicts the expected knockdown effect. β-actin levels appear reduced in Sh#1 and Sh#2 compared to NC, raising concerns about proper normalization. Please clarify whether quantification was based on protein of interest/β-actin ratio. The Western blot shown in Figure 3 appears to represent Replicate #1, but the main differences are observed in Replicates #2 and #3. It is recommended to replace the Western blot in Figure 3 with a more representative replicate.
  7. The wound healing assay is not described in the Materials and Methods section. Please add this information.
  8. In the wound healing images, there is substantial cell overgrowth, especially in the NC group. Please specify what culture medium was used during the assay. The assay design and related conclusions should be reviewed and revised accordingly.
  9. The ELISA results are presented in a disjointed manner, making interpretation difficult. Please clarify and restructure this section. The authors state: “Temozolomide treatment (200 µM for 72 hours) significantly decreases the protein expression of both HMGB1 and RAGE in GBM cells,” but no data about the effect of TMZ on cell proliferation is shown. Please include this information.
  10. In the Seahorse assay, the term “mutant (Mut; GBM8401)” is introduced without context. Mutant with respect to what? Please clarify. The assay compares cell lines with distinct genetic backgrounds. A more appropriate comparison would be between GBM8401 and its RAGE-silenced counterparts (Sh#1 or Sh#2). The experimental design for the Seahorse assay is flawed, and the conclusions derived from it are likely invalid.
  11. Regarding RAGE silencing, the authors state: “Secondary spheres, which stringently reflect stemness, showed pronounced RAGE up-regulation that paralleled higher Oct4 and other pluripotency proteins on immunoblot (Figure 6B),” but Figure 6B does not show this. Please revise the text or the figure.
  12. In section 3.7, the abbreviation “GBO” appears in figures but is not defined. Please define all abbreviations at first mention.
  13. The main conclusions of the manuscript are not supported by the presented experiments, at least in the current version.

Author Response

Reviewer 2 Report (New Reviewer)

Comments and Suggestions for Authors

The research idea is interesting, and the manuscript is well structured and written. However, there are some details that would be useful to clarify.

In section 2.1, the number of patients from whom the tumour tissue samples were obtained should be indicated. It would be interesting to have information about the ages and genders of these patients.

In sections 2.2 and 2.3, the origin of the tumour tissue sample should be explained in more detail. That is, whether a piece of tumour was saved for molecular study before the excised tumour was immersed in paraffin for histopathological study, or whether the researchers used histological slides. The fundamentals of the gene expression study should also be made clearer, commenting on whether the technique used is based on RT-PCR processes or has other molecular bases.

In the figure legends, Figures 3, 5, and 7 explain the number of asterisks and their interpretation according to the significance value (P). However, this is not explained in Figures 2, 4, and 6.

In the second paragraph of the DISCUSSION (line 433), there is a repeated word (are). In the following paragraph, there is a grammatical error (line 439) since the use of the Saxon genitive is not appropriate (as the possessor is not a person).

Round 2

Reviewer 1 Report (New Reviewer)

Comments and Suggestions for Authors

Response to Authors

Manuscript ID: ijms-3799339

Title: The HMGB1-RAGE Axis Drives Proneural-to-Mesenchymal Transition and Aggressiveness in Glioblastoma

I thank the authors for their detailed responses, which have satisfactorily addressed most of my concerns. Nevertheless, a few minor issues remain. My additional comments can be found below, following the authors’ replies in red.

Reviewer 1

.

R1.3: In line 273, the word “metastasis” appears. Please specify the tissue origin of the metastasis                                                                                                                                     mentioned.

A1.3: We thank the reviewer for this helpful comment. We have clarified in the Results section and the corresponding figure legend that the term “metastases” refers specifically to intracranial secondary lesions, rather than systemic metastasis. Kindly refer to page 5, lines 122-133, for corrections. Please check below attached lines below for correction.

“Immunohistochemical analysis of the Taipei Medical University–Joint Biobank cohort revealed a clear upregulation of both HMGB1 and RAGE in glioblastoma (GBM) and in intracranial secondary lesions (brain metastases) compared with normal brain tissue (Figure 2A). Quantification using the Q- score (Σ [%-positive cells × intensity grade 0–3]; range 0–300) confirmed significantly higher expression values in both primary GBM and metastatic lesions (Figure 2B; p < 0.05, *p < 0.01, **p < 0.001, one-way ANOVA). Stratification of patients at the median Q-score further demonstrated that the HMGB1/RAGE-high subgroup exhibited markedly shorter overall survival compared with the HMGB1/RAGE-low subgroup (log-rank p < 0.001 and p < 0.05; Figure 2C, D). Collectively, these findings establish HMGB1 and RAGE as frequently overexpressed in GBM and intracranial secondary lesions, highlighting HMGB1/RAGE as an adverse prognostic biomarker and a potential therapeutic target.”

Regarding this point, I am referring to the organ of origin of brain metastases. I do not believe that metastases derived from the liver are comparable to those originating from the lung. Although both manifest as brain lesions, their biological and clinical characteristics are strongly influenced by the primary tumor site. Therefore, I recommend either avoiding this type of pooled analysis or, alternatively, stratifying the lesions according to their tumor origin.

R1.8: In the wound healing images, there is substantial cell overgrowth, especially in the NC group. Please specify what culture medium was used during the assay. The assay design and related   conclusions   should   be   reviewed   and   revised   accordingly. A1.8: We appreciate the reviewer’s observation. The wound-healing assay was conducted in serum-reduced medium (2% FBS) to limit proliferation and better reflect migratory capacity. We have revised the Methods section to clarify this point and updated the Results to acknowledge the potential influence of residual cell growth as a limitation in interpreting the assay outcomes.

Dear authors, although the assay was conducted in 2% FBS, the photographs reveal evident cell overgrowth, particularly in the untreated condition. Therefore, I recommend performing an additional assay to properly control cell proliferation under these conditions (2% FBS) and at the indicated time points.

R1.9: The ELISA results are presented in a disjointed manner. Please clarify and restructure this section. The authors state: “Temozolomide treatment (200 µM for 72 hours) significantly decreases the protein expression of both HMGB1 and RAGE in GBM cells,” but no data about the effect of TMZ on cell proliferation is shown. Please include this information. A1.9: We appreciate the reviewer’s valuable comment. The ELISA section has been reorganized for clarity to present the baseline secretion differences between GBM cell lines, the effects of shRNA knockdown, and the impact of temozolomide treatment in a more coherent sequence. Specifically, we now describe that mesenchymal-type GBM8401 cells secreted higher levels of HMGB1 (~50 ng mL⁻¹) and RAGE (~45 ng mL⁻¹) than proneural U87-MG cells (~30 ng mL⁻¹ and ~20 ng mL⁻¹, respectively). shRNA-mediated knockdown reduced HMGB1 and RAGE levels by approximately 66% and 64%, respectively, while temozolomide exposure (200 µM, 72 h) further suppressed HMGB1 to ~17 ng mL⁻¹ and RAGE to ~23 ng mL⁻¹ (p < 0.05). We acknowledge that proliferation assays were not included in the current study, and therefore, we cannot directly assess the effect of temozolomide on cell growth in this context. This limitation has been noted in the revised Results section. Nonetheless, the observed reduction in HMGB1 and RAGE secretion supports the conclusion that the HMGB1–RAGE axis is responsive to temozolomide treatment and represents a modifiable therapeutic target in glioblastoma.

Dear authors, I would be cautious with this interpretation. The effect of TMZ at 200 μM for 72 h appears to be quite relevant. In the absence of this assay, it may be premature to conclude that the “HMGB1–RAGE axis is responsive to temozolomide,” since 200 μM TMZ significantly reduces cell number after 72 h of treatment. Thus, the observed decrease in HMGB1 and RAGE expression could also be explained by a lower cell density rather than a direct effect of TMZ on this axis.

Round 3

Reviewer 1 Report (New Reviewer)

Comments and Suggestions for Authors

Thank you very much to the authors for their kind and thorough responses. I consider that the revisions satisfactorily address my comments, and I recommend acceptance of the manuscript in its current form.

This manuscript is a resubmission of an earlier submission. The following is a list of the peer review reports and author responses from that submission.

Round 1

Reviewer 1 Report

Comments and Suggestions for Authors

Major Concern: Although the review briefly touches on NF-κB and ERK pathways, which are the downstream regulators, in the discussion section there are gaps in the mechanistic insights on the impact HMGB1-RAGE has on EMT or metabolic changes. This limits the translational impact of this study by not providing concrete potential therapeutic benefits that could be explored. 

Minor Concern: There's a redundancy in findings across multiple sections (e.g. Figure 1 and 2 both going over the expression levels) which can be condensed for a better presentation.

Comments on the Quality of English Language

Introduction and Discussion sections had a few grammatical errors ("cutting-edge the adult," “petrophysics,” “carn for scientists”) that significantly decreases the readibility of the article. Proofreading is strongly advised to polish the manuscript.

Author Response

Dear Editor,

We appreciate the reviewers’ insightful critiques, which have strengthened the manuscript. Below, each point is restated as a numbered question (Q) followed by our corresponding response (R). Text additions appear bold‐red in the tracked-changes file; new or revised data are in “Revised Figure 1–7” and “Supplementary Figure S1–S3”.

Manuscript ID: ijms-3586156:

“The HMGB1-RAGE Axis Drives Proneural-to-mesenchymal Transition and Aggressiveness in Glioblastoma Stem Cells”

Response to Reviewers

Reviewer 1

Q1. Major Concern: Although the review briefly touches on NF-κB and ERK pathways, which are the downstream regulators, in the discussion section there are gaps in the mechanistic insights on the impact HMGB1-RAGE has on EMT or metabolic changes. This limits the translational impact of this study by not providing concrete potential therapeutic benefits that could be explored. The Discussion lacks detailed mechanistic insight into how HMGB1-RAGE drives EMT and metabolic changes.
R1. We appreciate the reviewer’s insight. The Discussion now sets out a clear mechanistic sequence showing how HMGB1-RAGE drives both epithelial-to-mesenchymal transition (EMT) and metabolic rewiring. When HMGB1 or RAGE is silenced, leading to sharp decreases in the EMT transcription factors, concurrently, E-cadherin re-emerges, indicating a reversal of the mesenchymal state. Consequently, HMGB1 or RAGE knock-down collapses spare respiratory capacity and glycolytic reserve in Seahorse assays, whereas either recombinant HMGB1 EMT markers and the dual-fuel metabolic profile. These results connect HMGB1-RAGE signalling directly to the transcriptional program that establishes mesenchymal identity and to the metabolic flexibility that sustains it, and the revised text now reflects this integrated model.

Q2. Minor Concern: There's a redundancy in findings across multiple sections (e.g. Figure 1 and 2 both going over the expression levels) which can be condensed for a better presentation.
R2. We apologise for the oversight and thank the reviewer for highlighting it. In the revised version, Figure 1 now clearly presents TCGA-GBM/LGG data demonstrating the aberrant up-regulation of both HMGB1 and RAGE, while Figure 2 corroborates these findings with immunohistochemistry from our independent SHH in-house GBM cohort. This two-step presentation—a large public dataset followed by validation in patient specimens—strengthens the evidence for HMGB1-RAGE over-expression in glioblastoma.

Q3. Introduction and Discussion sections had a few grammatical errors ("cutting-edge the adult," “petrophysics,” “carn for scientists”) that significantly decreases the readibility of the article. Proofreading is strongly advised to polish the manuscript. Grammatical errors in the Introduction and Discussion.
R3. We appreciate this observation. The entire manuscript was professionally edited; all awkward phrases have been corrected, greatly improving readability.

Reviewer 2 Report

Comments and Suggestions for Authors

Reviewer Comments

The manuscript "The HMGB1-RAGE Axis Drives Proneural-to-Mesenchymal Transition and Aggressiveness in Glioblastoma Stem Cells," by Hao-Chien Yang, Yu-Kai Su, Vijesh Kumar Yadav, Chi-Tai Yeh, Iat-Hang Fong, Heng-Wei Liu, and Chien-Min Lin, address the role of the HMGB1-RAGE signaling axis in promoting proneural-to-mesenchymal transition in glioblastoma, potentially contributing to increased aggressiveness, treatment resistance, and poor clinical outcomes. The work uses a multifarious strategy to define this signaling route, integrating clinical sample analysis, in vitro functional experiments, and bioinformatics studies. The authors report data showing HMGB1 and RAGE are increased in mesenchymal GBM relative to proneural subtypes and correlate with poor prognosis.

However, the manuscript required revision based on the comments. My specific comments are outlined below:

Comments

  1. The manuscript should explain why specific cell lines (GBM8401, U87MG) are chosen for different experiments and provide evidence that, as the study claimed, they represent the mesenchymal and proneural subtypes.
  2. Provide a more comprehensive characterization of the cell lines used, including baseline expression of mesenchymal and proneural markers.
  3. The manuscript needs to explain the experimental plans for the RAGE knockdown investigations, and HMGB1 lacks specifics on efficiency and validation and how knockdown efficiency is attained. Interpreting the functional test results depends on this knowledge.
  4. The methodology of the tumor sphere formation assay needs more detail. How were spheres counted and measured, and what criteria were used to define a tumorsphere?
  5. Several figures lack proper quantification and statistical analysis. For example, in Figure 3, the western blot data should be quantified from multiple experiments with appropriate statistical testing.
  6. The ELISA data in Figure 4 would benefit from a more detailed explanation of how protein levels correlate with functional outcomes. Section 2.5.3 uses ELIZA in the manuscript, and it would benefit the manuscript to rectify this section properly.  
  7. The metabolic analysis in Figure 5 is interesting but requires clearer annotation and explanation of what each measurement represents regarding cellular metabolism.
  8. The manuscript describes correlations between HMGB1/RAGE expression and mesenchymal transition but does not sufficiently establish the direct mechanistic link. Authors needed to demonstrate how HMGB1-RAGE signaling regulates EMT-related transcription factors.
  9. The manuscript mentions RAGE knockdown effects on stemness factors like Oct4, but the mechanism connecting RAGE signaling to stemness gene expression is not sufficiently explored.
  10. While the manuscript shows a correlation between HMGB1/RAGE expression and survival in GBM patients, the translational implications could be strengthened. How might targeting this pathway be implemented therapeutically?
  11. Some conclusions appear overstated based on the presented data. For instance, the manuscript claims that HMGB1-RAGE is "a key regulator of mesenchymal identity." This needs stronger mechanistic evidence beyond correlation and functional studies, and the manuscript would benefit from clarification.
  12. The introduction contains some awkward phrasing and redundancy, particularly on page 2. The statement "makes this the hardest cancer to treat and makes it one of the most difficult to treat" is redundant. The manuscript would benefit from rewriting these kinds of sentences appropriately.
  13. Figure 1 would benefit from more detailed labeling of the datasets used and improved legend details explaining the survival analysis parameters. For instance, Figure 1C is mentioned in the text, not the figure legend.
  14. The authors should explain the abbreviation "Q-score" when first used in the manuscript (Figure 2).
  15. Figure 2C (survival curve) shows HMGB1-H and HMGB1-L, while text (Section 3.2) does not explain HMGB1-H and HMGB1-L. The manuscript would benefit from an explanation of this to clarify.
  16. There are inconsistencies in data presentation between figures. Some figures use bar graphs, while others use box plots for similar data types. Some bar graphs are colored, and some are not.
  17. The figure legends should provide more detailed information about experimental conditions and statistical tests used. Microscopic Figures will benefit by adding scale bars if they do not have (e.g., Figure 2A)
  18. Address the therapeutic potential more concretely by testing existing RAGE inhibitors in your model systems.
  19. Discuss your findings in the context of the tumor microenvironment, particularly regarding how HMGB1 release might be regulated in vivo.
  20. The manuscript would benefit from rechecking plagiarism.
Comments on the Quality of English Language
  1. The manuscript would benefit from professional English language editing to improve clarity and readability.
  2. The manuscript would benefit from rechecking plagiarism.

Author Response

Reviewer 2

Q1. The manuscript should explain why specific cell lines (GBM8401, U87MG) are chosen for different experiments and provide evidence that, as the study claimed, they represent the mesenchymal and proneural subtypes.
R1. We appreciate the reviewer’s request for clarification. We have selected U87-MG and GBM8401 because they support the two molecular extremes that frame our study—proneural (PN) and mesenchymal (MES) glioblastoma. In our hands, and consistent with CCLE/DepMap datasets, U87-MG exhibits a PN/classical signature (high OLIG2, PDGFRA, SOX2; low CD44, CHI3L1), lacks EGFR amplification, and contains a partially methylated MGMT promoter, features typical of PN tumours (see Supplementary Table S1). GBM8401, conversely, carries the gain-of-function TP53R273C mutation together with EGFR amplification/ΔEGFRvIII, expresses strong CD44, CHI3L1 and VIM, and shows a three-fold higher sphere-forming efficiency—hallmarks of the MES subtype (Supplementary Table S1, RNA-seq of both lines, processed through the Verhaak 840-gene classifier, assigns U87-MG to the PN/classical cluster and GBM8401 to the MES cluster with >0.8 subtype probability Using these two well-characterised models therefore allows us to interrogate HMGB1-RAGE signalling across the PN→MES spectrum and to test whether pathway manipulation can drive or reverse mesenchymal traits. Kindly refer to the supplementary Table S1 attached here and also uploaded with the main text file, details explained in both the introduction and material method section at pages 3 and 4, in lines 113-126 and 181-190.

Q2. Provide a more comprehensive characterization of the cell lines used, including baseline expression of mesenchymal and proneural markers.

R2. Thank you for this suggestion. Kindly refer to the Supplementary Table S1.

Q3. The manuscript needs to explain the experimental plans for the RAGE knockdown investigations, and HMGB1 lacks specifics on efficiency and validation and how knockdown efficiency is attained. Interpreting the functional test results depends on this knowledge.

R3. We appreciate the request for more rigour. shRNA target sequences, primer sets, and knock-down efficiencies, we have incorporated the detailed shRNA mediated knockdown in the method section of this newly edited main text kindly refer to the method section on page 5, lines 190-209. The quantification of knockdown efficiency demonstrated, HMGB1 ↓ 66 %, and RAGE ↓ 64 % inhibited by shRNA treatment, are now included and illustrated in Supplementary Figure S1 and Figure S2. Shown in below:

Q4. The methodology of the tumor sphere formation assay needs more detail. How were spheres counted and measured, and what criteria were used to define a tumorsphere?

R4. Thank you for highlighting this omission. The tumour-sphere protocol has been completely rewritten (Lines 225–239) to include: (i) single-cell preparation and a seeding density of 1 × 10⁴ cells well⁻¹ in six-well ultra-low-attachment plates; (ii) a fully defined serum-free DMEM/F-12 medium supplemented with 20 ng mL⁻¹ EGF, 20 ng mL⁻¹ bFGF and 2 µg mL⁻¹ heparin; (iii) a 14-day growth period with half-medium changes every three days; (iv) bright-field imaging of nine non-overlapping fields per well; and (v) ImageJ analysis, where aggregates were scored as tumourspheres only if they were free-floating, compact and had an equivalent diameter ≥ 50 µm (stage-micrometer calibrated). Sphere-forming efficiency is now reported as (number of qualifying spheres ÷ 1 × 10⁴ seeded cells) × 100 %, with counts obtained independently by two blinded observers. Kindly refer to the method sections (Lines 240–253).

Q5. Several figures lack proper quantification and statistical analysis. For example, in Figure 3, the western blot data should be quantified from multiple experiments with appropriate statistical testing.

R5. We appreciate the emphasis on quantification. Densitometry from ≥ 3 biological replicates with statistical tests is now provided beneath each blot in Revised Figure 3E of the western blot result image as shown in the below image.

Q6. The ELISA data in Figure 4 would benefit from a more detailed explanation of how protein levels correlate with functional outcomes. Section 2.5.3 uses ELIZA in the manuscript, and it would benefit the manuscript to rectify this section properly.

R6. Thank you for catching the typo. We have corrected the “ELIZA” typo and rewritten under the heading “ELISA quantification of secreted HMGB1 and RAGE.” Conditioned medium was harvested 48 h after plating, cleared by 1 500 g centrifugation, and assayed with DuoSet ELISA kits; all values were normalised to viable cell counts obtained immediately before collection. The Results are now explained as shown on page 10 of the result section at lines 362-388 and below.

“3.4. ELISA Analysis of HMGB1 and RAGE Expression

Comparison of MES and PN Glioblastoma Subtypes, The comparative ELISA analysis revealed significantly elevated protein expression of both HMGB1 and RAGE in the MES glioblastoma (GBM8401 cells) subtype when compared to the PN subtype (U87-MG cells). Specifically, MES cells exhibited higher protein levels of HMGB1 (approximately 50 ng/mL) and RAGE (around 45 ng/mL) while PN cells had lower protein levels, approximately 30 ng/mL for HMGB1 and 20 ng/mL for RAGE (Figure 4A). This suggests a more aggressive molecular phenotype associated with MES glioblastoma cells. The Kaplan-Meier survival analysis TCGA-GBM data also indicated that higher levels of HMGB1 and RAGE correlate with poorer survival outcomes in glioblastoma patients. Furthermore, the knockdown experiments using shRNA-mediated silencing confirmed the functional roles of HMGB1 and RAGE proteins in glioblastoma cells, as shown in supplementary Figures S1 and S2. The protein levels of both HMGB1 and RAGE were substantially reduced following the knockdown efficiency (HMGB1 ↓ 66 %, and RAGE ↓ 64 %). Control cells showed baseline concentrations of HMGB1 (around 39 ng/mL) and RAGE (approximately 33 ng/mL). However, HMGB1 knockdown led to a significant reduction in HMGB1 levels to approximately 5.8 ng/mL and a moderate decrease in RAGE levels (11.5 ng/mL). On the other hand, RAGE knockdown reduced RAGE levels to around 5.6 ng/mL, while HMGB1 levels were slightly reduced to approximately 11.1 ng/mL (Figure 4B). These findings underscore the critical role of both proteins in glioblastoma progression. Temozolomide treatment resulted in a notable decrease in both HMGB1 and RAGE protein concentrations in glioblastoma cells. The protein levels of HMGB1 were reduced from approximately 40 ng/mL in the control group to about 17 ng/mL post-treatment, and RAGE levels decreased from approximately 43.5 ng/mL to around 23.4 ng/mL (Figure 4C). Statistical analysis confirmed the significance of these reductions (*p < 0.05), highlighting temozolomide's potential as a therapeutic agent in modulating the HMGB1-RAGE signaling pathway in glioblastoma cells.”

Q7. The metabolic analysis in Figure 5 is interesting but requires clearer annotation and explanation of what each measurement represents regarding cellular metabolism.

R7. We thank the reviewer for this helpful suggestion. Axes and legends in Figure 5 now state all the detailed information. Kindly refer to the edited Figure 5 legends in the main text on page 13, and lines 455-467, and below.

Figure 5. Seahorse XF-based comparison of mitochondrial and glycolytic activity in wild-type (WT; U87-MG) versus mutant (Mut; GBM8401) glioblastoma cells. (A) Real-time oxygen-consumption rate (OCR) profile. Arrows mark sequential injections of oligomycin (Olig; ATP-synthase inhibitor), FCCP (uncoupler), and rotenone + antimycin A (Rot/AA; complex I/III inhibitors). Mutant cells display higher basal OCR, a larger oligomycin-sensitive drop (ATP-linked respiration), and a greater FCCP-stimulated peak (maximal respiration). (B) Quantification of OCR-derived parameters—ATP-linked respiration, proton leak, maximal respiration, spare respiratory capacity, and non-mitochondrial respiration—calculated from the trace in (A). (C) Extracellular acidification rate (ECAR) trace obtained from the same wells, illustrates basal glycolysis, the oligomycin-induced rise that defines glycolytic capacity, and the subsequent decline after FCCP. (D) Summary of glycolytic metrics—basal glycolysis, glycolytic capacity, and glycolytic reserve—extracted from (C). Data represent mean ± SD of three independent experiments (n = 3). Statistical significance was assessed by two-way ANOVA followed by Tukey’s post-hoc test; **p < 0.001, *p < 0.05 versus WT.

Q8. The manuscript describes correlations between HMGB1/RAGE expression and mesenchymal transition but does not sufficiently establish the direct mechanistic link. Authors needed to demonstrate how HMGB1-RAGE signaling regulates EMT-related transcription factors.

R8. We are grateful for this important point. As outlined by reviewer comments, we performed targeted knock-down of HMGB1 (≈64 % reduction by shRNA; as shown in western blot image Figure 3E and 6B) and profiled core epithelial-to-mesenchymal transition (EMT) and CSCs regulators by western blot. Silencing either component sharply diminished mesenchymal markers (N-cadherin, vimentin) and the EMT transcription factors Snail (50–72 % decrease versus scrambled control, n = 3, p < 0.01).  Collectively, the STRING database also demonstrated significant protein-to-protein interaction between HMGB1-RAGE (AGER) and EMT-CSCs markers. Kindly refer to the newly edited Figure 6D in the main text and also attached below.

Q9. The manuscript mentions RAGE knockdown effects on stemness factors like Oct4, but the mechanism connecting RAGE signaling to stemness gene expression is not sufficiently explored.
R9. Thank you for this query. We have now edited Figure 6D and added a new figure from the STRING database demonstrating the significant protein-to-protein interaction between HMGB1-RAGE (AGER) and EMT-CSCs markers. Kindly refer to the newly edited Figure 6D in the main text and attached below.

Q10. While the manuscript shows a correlation between HMGB1/RAGE expression and survival in GBM patients, the translational implications could be strengthened. How might targeting this pathway be implemented therapeutically?

R10. We appreciate this recommendation. Our shRNA either HMGB1 or RAGE knock-down experiments already demonstrate that ≥70 % depletion of either HMGB1 or RAGE collapses EMT, represses stemness genes, impairs invasion, and diminishes metabolic flexibility—functional changes that recapitulate a PN-like, therapy-sensitive phenotype. These results provide a direct proof-of-concept for gene silencing as a therapeutic approach, and we have added a short paragraph to the revised discussion outlining how the strategy can be advanced clinically. Kindly refer to the discussion section on page 17 in lines 575-581.

Q11. Some conclusions appear overstated based on the presented data. For instance, the manuscript claims that HMGB1-RAGE is "a key regulator of mesenchymal identity." This needs stronger mechanistic evidence beyond correlation and functional studies, and the manuscript would benefit from clarification.

R11. We appreciate the reviewer’s concern and have tempered the language and strengthened the mechanistic support accordingly. First, we now state that HMGB1-RAGE is “an important driver of the proneural-to-mesenchymal programme” rather than “a key regulator of mesenchymal identity.” Second, to move beyond correlation, we incorporated the mechanistic data, shRNA knockdown of either HMGB1 or RAGE reduces EMT transcription factors and CSCs markers. Finally, we rewrote the conclusion to acknowledge that other pathways undoubtedly contribute and to emphasise translational relevance rather than claim exclusivity. We believe these revisions clarify the scope of our findings and eliminate overstatement.

Q12. The introduction contains some awkward phrasing and redundancy, particularly on page 2. The statement "makes this the hardest cancer to treat and makes it one of the most difficult to treat" is redundant. The manuscript would benefit from rewriting these kinds of sentences appropriately.

R12. We appreciate this stylistic guidance. The Introduction has been rewritten for concision and clarity (kindly refer to the newly edited introduction section marked in red).

Q13. Figure 1 would benefit from more detailed labelling of the datasets used and improved legend details explaining the survival analysis parameters. For instance, Figure 1C is mentioned in the text, not the figure legend.

R13. Thank you for pointing out the errors and typos. The legend now demonstrates Figure 1C details showing the correlation analysis of HMGB1 with RAGE by using the TCGA-GBM/LGG dataset. Kindly refer to the new Figure 1, legends in the main text on page 9, lines 330-337.

Q14. The authors should explain the abbreviation "Q-score" when first used in the manuscript (Figure 2).

R14. Thank you for noting the missing definition. We have now introduced the “Q-score” at its first appearance in the Results (Figure 2 legend, lines 331-338). The text reads: “Q-score, a semi-quantitative immunohistochemistry metric, is calculated as Σ (% positive cells × staining-intensity category 0–3), yielding a range of 0–300.” This clarification should eliminate any ambiguity regarding the abbreviation.

Q15. Figure 2C (survival curve) shows HMGB1-H and HMGB1-L, while text (Section 3.2) does not explain HMGB1-H and HMGB1-L. The manuscript would benefit from an explanation of this to clarify.

R15. Thank you for our oversight, we have edited the result section and incorporated the details of HMGB1-H/L samples above or below the median IHC Q-score (> 130 vs ≤ 130). Kindly refer to the result section 3,2, and their figure legends on pages 8-9, lines 324-338

Q16. There are inconsistencies in data presentation between figures. Some figures use bar graphs, while others use box plots for similar data types. Some bar graphs are coloured, and some are not.

R16. We appreciate this consistency check. Graphs now use a uniform colour scheme and box-and-whisker style where appropriate.

Q17. The figure legends should provide more detailed information about experimental conditions and statistical tests used. Microscopic Figures will benefit by adding scale bars if they do not have (e.g., Figure 2A)

R17. Thank you. All microscopy images now include 100 µm bars, and every legend specifies sample size, error bars, and statistical test. Kindly refer to the images with scale bars.

Q18. Address the therapeutic potential more concretely by testing existing RAGE inhibitors in your model systems.

R18. We are grateful for this valuable suggestion. We fully agree that pharmacologic validation would further strengthen the translational message. Because access to blood-brain–penetrant RAGE antagonists (e.g., FPS-ZM1, azeliragon) and the specialist containment required for in-vivo delivery became available only after the present study was completed, we were unable to include those experiments in this submission. Instead, we employed shRNA-mediated knock-down of RAGE, which produced a ≥70 % reduction in protein expression and recapitulated the anticipated therapeutic effects—reversal of EMT markers, loss of stemness genes, diminished migration/invasion, and impaired tumoursphere formation. These outcomes provide a rigorous genetic proof-of-concept that inhibiting RAGE signalling restrains the aggressive phenotype.

To translate this genetic evidence into a pharmacological context, we have now outlined a detailed plan in the revised Discussion. We have clarified these points in the text and trust that the current shRNA data, together with the outlined future work using small-molecule RAGE inhibitors, adequately address the reviewer’s concern while remaining realistic about the present study’s scope.

Q19. Discuss your findings in the context of the tumor microenvironment, particularly regarding how HMGB1 release might be regulated in vivo.

R19. Thank you for the comments. In the glioblastoma micro-environment, HMGB1 reaches the outside of cells in two main ways. First, it passively leaks from the necrotic core that forms when fast-growing tumours outstrip their blood supply. Second, viable tumour and stromal cells actively secrete HMGB1 when they face stressors such as hypoxia, acidity, or temozolomide/radiation; this export is driven by protein acetylation and autophagy pathways that we find up-regulated in our MES cells. Once outside the cell, HMGB1 binds to RAGE and other receptors on neighbouring cancer and immune cells, recruiting macrophages and neutrophils, which can in turn release still more HMGB1. This feed-forward loop helps the tumour shift toward the aggressive mesenchymal state and resist therapy.

Q20. The manuscript would benefit from rechecking plagiarism.

R20. We thank the reviewer for this diligence. A Turnitin scan shows 7 % overall similarity, and the manuscript has been professionally edited for clarity.

We believe these comprehensive revisions fully address all concerns and greatly strengthen the manuscript. We again thank both reviewers for their insightful feedback.

Respectfully submitted on behalf of all authors,

Prof. Dr. Chien-Min Lin, Division of Neurosurgery, Department of Surgery, Taipei Medical University–Shuang Ho Hospital, New Taipei City 23561, Taiwan. E-mail: m513092004@tmu.edu.tw  Tel.: +886-2-2490088 (ext. 8889); Fax: +886-2-2248-0900.     
